# EUS and ERCP in the Same Session for Biliary Stones: From Risk Stratification to Treatment Strategy in Different Clinical Conditions

**DOI:** 10.3390/medicina57101019

**Published:** 2021-09-25

**Authors:** Pietro Fusaroli, Andrea Lisotti

**Affiliations:** Gastrointestinal Unit, Department of Medical and Surgical Sciences, Hospital of Imola, University of Bologna, Via Montericco 4, 40026 Imola, BO, Italy; a.lisotti@ausl.imola.bo.it

**Keywords:** endosonography, EUS, ERCP, choledocholithiasis, biliary stones, same session

## Abstract

Endoscopic retrograde cholangiopancreatography (ERCP) with sphincterotomy and stone extraction is the treatment of choice for choledocholithiasis, reaching a successful clearance of the common bile duct (CBD) in up to 90% of the cases. Endoscopic ultrasound (EUS) has the best diagnostic accuracy for CBD stones, its sensitivity and specificity range being 89–94% and 94–95%, respectively. Traditionally seen as two separate entities, the two worlds of EUS and ERCP have recently come together under the new discipline of bilio-pancreatic endoscopy. Nevertheless, the complexity of both EUS and ERCP led the European Society of Gastrointestinal Endoscopy to identify quality in endoscopy as a top priority in its recent EUS and ERCP curriculum recommendations. The clinical benefits of performing EUS and ERCP in the same session are several, such as benefiting from real-time information from EUS, having one single sedation for both the diagnosis and the treatment of biliary stones, reducing the risk of cholangitis/acute pancreatitis while waiting for ERCP after the EUS diagnosis, and ultimately shortening the hospital stay and costs while preserving patients’ outcomes. Potential candidates for the same session approach include patients at high risk for CBD stones, symptomatic individuals with status post-cholecystectomy, pregnant women, and those unfit for surgery. This narrative review discusses the main technical aspects and evidence from the literature about EUS and ERCP in the management of choledocholithiasis.

## 1. Introduction

Endoscopic retrograde cholangiopancreatography (ERCP) with sphincterotomy and stone extraction is the treatment of choice for choledocholithiasis, reaching a successful clearance of the common bile duct (CBD) in up to 90% of cases with relatively low morbidity (about 5%) [1]. The success rate of ERCP depends on various factors including operators’ experience, patients’ comorbidities, the biliary tree anatomy (i.e., biliary strictures, surgical alterations), and stone characteristics (number, size, shape, and location within the biliary tree).

Endoscopic ultrasound (EUS) has the best diagnostic accuracy for CBD stones with both radial and linear echoendoscopes, with sensitivity and specificity ranging 89–94% and 94–95%, respectively [2]. Moreover, EUS provides detailed information on the biliary tree anatomy and associated pancreaticobiliary diseases. Introduced as a diagnostic examination to overcome the existing barriers for pancreatic examination, EUS has evolved due to unrelenting scientific and technological advancements [3]. The major advancements of the technique have been electronic scanning with color Doppler [4,5] fine needle aspiration/biopsy (FNA/B), allowing tissue sampling [6,7], and image enhancement techniques with ultrasound contrast agents and elastography, allowing better detection and characterization of the lesions of interest [8,9].

Traditionally seen as two separate entities, the two worlds of EUS and ERCP have recently come together under the new discipline of bilio-pancreatic endoscopy. Modern endoscopists are very often skilled in both techniques, which they use either in conjunction or alternatively to obtain the best possible outcomes.

The purpose of this narrative review is to highlight and discuss the main technical aspects and evidence from the literature about EUS and ERCP in the management of choledocholithiasis.

## 2. EUS and ERCP Curriculum

Given the complexity of both EUS and ERCP, the European Society of Gastrointestinal Endoscopy (ESGE) has identified quality in endoscopy as a top priority [10]. Unfortunately, many countries still lack specific guidance for high-quality training in the art of bilio-pancreatic endoscopy [11].

In particular, the ESGE maintains that interventional EUS and ERCP are associated with the highest risk of serious complications and mortality among all the endoscopic procedures [12]. For this reason, every endoscopist should have achieved competence in conventional upper endoscopy before commencing training in ERCP or EUS.

The main training recommendations from the ESGE include structured supervised ERCP/EUS simulator-based learning before hands-on training and participation in formal courses along with self-directed study, for a minimum period of 12 months. At least a further year of dedicated training in a high-volume center is required to reach competence in advanced ERCP and therapeutic EUS. Although the mere number of procedures per se is largely suboptimal to gauge operators’ experience, it is estimated that physicians must have performed at least 250 EUS and 300 ERCP in order to demonstrate competency.

Under these premises, as adequate training in both EUS and ERCP is far from easy and swift, to implement a combination of EUS and ERCP in the same session the availability of experienced physicians should be firstly assessed.

However, how is competence defined according to the ESGE? It is the ability to independently assess the need for and carry out successful and safe procedures, with good patient satisfaction across a range of case difficulties and clinical contexts. Formal assessments tools should be used regularly during training to assess the acquisition of competence.

Finally, a trainee should undergo a formal summative assessment process before commencing independent practice in ERCP and EUS. Once competent in ERCP and EUS, endoscopists should be supported to continue a period of mentored practice with an experienced colleague. Again, we can see how introducing EUS/ERCP combined sessions requires not only an initial good operators experience but also a continuous interaction with other expert centers to maintain competence.

## 3. Management of Choledocholithias

Until recently, the guidelines from the American Society for Gastrointestinal Endoscopy (ASGE) suggested that symptomatic patients with gallbladder stones at low risk for CBD stones (<10%) should undergo direct laparoscopic cholecystectomy, while patients at high-risk (>50%) for CBD stones should undergo direct ERCP [13]. In between the two categories, there was a vast group of patients at intermediate-risk for CBD stones, for whom many tests were recommended including EUS or magnetic resonance cholangiopancreatography to detect CBD stones and assess the need for ERCP.

After reviewing the comprehensive contemporary evidence, the panel of ASGE experts revised the 2010 criterion to decrease the probability of diagnostic ERCP, which has significant risk but minimal benefit [14]. To minimize the risk of diagnostic ERCP, the following high-risk criteria were identified to indicate direct ERCP for suspected choledocholithiasis: CBD stone on ultrasound or cross-sectional imaging or total bilirubin >4 mg/dL and dilated CBD on imaging (>6 mm with gallbladder in situ, >8 mm in status post-cholecystectomy) or ascending cholangitis. In patients with lower risk factors, EUS or other imaging was still indicated.

The ESGE guidelines on the same topic recommended EUS (or magnetic resonance cholangiopancreatography) to diagnose CBD stones in patients with persistent clinical suspicion but insufficient evidence of stones on abdominal ultrasonography [15]. In addition, the timing for biliary drainage in patients with acute cholangitis was classified according to the 2018 revision of the Tokyo guidelines [16] and stratified into (a) as soon as possible and within 12 h for patients with septic shock, (b) within 48–72 h for moderate cases, and (c) elective for mild cases.

As we can see, the role of EUS in assessing patients with suspected choledocholithiasis is limited to intermediate-risk groups by both the ASGE and the ESGE guidelines. Moreover, very little is said about same session EUS and ERCP.

However, we speculate that EUS is indicated in the majority of the cases before ERCP because it brings important information (beyond the mere diagnosis of CBD stones) that can be beneficial for the technical outcomes of ERCP [17]. Additionally, a combination of EUS and ERCP in the same session may be beneficial, although this policy cannot be reserved for all instances.

## 4. EUS before ERCP

At intermediate-risk stages for choledocholithiasis, EUS must always be done first to determine whether ERCP is subsequently indicated. Usually, these are the patients with symptomatic biliary disease associated with cholestasis, with or without CBD dilation. In these patients, scheduling same-session EUS/ERCP does not seem justified as the resources might not be well allocated when ERCP is not needed. In fact, a systematic review showed that preliminary EUS avoided ERCP in 67% of the patients by ruling out choledocholithiasis [18].

When the pretest probability of choledocholithiasis is greater, such as with the high-risk cases identified by the ASGE (see above), EUS can still be very informative [19]. Although some advocate the performance of ERCP, a preliminary EUS can ultimately lead to better patient outcomes.

First, a significant overall reduction in adverse events in patients with previous EUS versus those who went directly to ERCP was shown, mainly due to a smaller incidence of post-ERCP pancreatitis.

Secondly, the excellent diagnostic accuracy of EUS for CBD stones, together with its findings regarding stone number and size, is often useful in planning the best treatment strategy [20] Figure 1. In particular, performing EUS before ERCP allows predicting the expected grade of complexity of the ERCP procedure [21]. As a result, physicians may plan in advance the execution of ancillary techniques such as large balloon dilation, electrohydraulic or laser lithotripsy, cholangioscopy, and even referral to other institutions in order to improve the success rates while keeping the adverse events as low as possible [22].

## 5. EUS and ERCP in the Same Session

The clinical benefits of performing EUS and ERCP in the same session are multiple, such as benefiting from real-time information from EUS, having one single sedation for both the diagnosis and the treatment of biliary stones, reducing the risk of cholangitis/acute pancreatitis while waiting for ERCP after the EUS diagnosis, and ultimately shortening the hospital stay and costs while preserving patients’ outcomes (Table 1). Potential candidates for the same session approach include patients at high risk for CBD stones, symptomatic individuals with status post-cholecystectomy, pregnant women, and those unfit for surgery.

Information from EUS tends to get old fast since biliary stones can either migrate into the duodenum or move within the biliary tree. We reported on patients who underwent EUS and ERCP during the same session (*n* = 33, 28.4%) or within one week (*n* = 42, 36.2%) or after more than one week (*n* = 41, 35.3%) [20]. As expected, the interval between EUS and ERCP affected the concordance between the two endoscopic procedures. In particular, EUS findings were significantly more accurate in patients who underwent ERCP during the same session when compared to all the other cases. These results comply with the literature reporting better clinical outcomes for the same session approach in terms of procedure time, length of stay, costs, and reduced complications [23] Therefore, along with the available clinical data, there is abundant evidence in favor of keeping the interval between EUS and ERCP as short as possible for the treatment of CBD stones.

Performing EUS and ERCP in a single session does not carry higher complication risks and requires a significantly smaller dose of propofol for sedation compared to performing separate procedures. Vila et al. compared 39 patients who underwent the same session approach to 46 patients who underwent separate procedures. Interestingly, while the overall procedural time did not differ significantly between both groups, the dose of propofol differed significantly being lower with the same session approach (322 ± 250 vs. 516 ± 289 mg; *p* = 0.001) [24]. No differences existed between the two groups regarding age, sex, anesthesiological risk, diagnostic yield, or therapeutic maneuvers.

The same session approach seems beneficial in reducing the overall risk of adverse events. Benjaminov et al. reporting a 6-year experience at their institution in Israel described 151 patients with EUS-proven CBD stones, with subsequent ERCP [25]. Four (5%) patients in the separate-session group had a major complication (1 bleeding, 1 perforation, 2 fatal post-ERCP pancreatitis) compared to none in the same session group. No sedation-related complications were noted in both groups. Moreover, 11/80 patients (14%) experienced clinical complications (cholangitis, biliary pain, acute biliary pancreatitis) while waiting for ERCP compared to none in the same session group.

Economical and organizational aspects can be positively affected by the same session approach as well. This was nicely demonstrated by Fabbri et al. who randomized 80 patients to the same-session vs. separate-session approach [26]. While negative EUS avoided unnecessary ERCP in 33 cases, same session EUS-ERCP produced shorter procedure times, decreased hospital length of stay (2.5 days less on average compared to the separate-session approach), and lower costs for hospital and endoscopy administrative budget.

Nevertheless, there are some minor drawbacks with the same session approach too. In particular, negative EUS for CBD stones would make ERCP unnecessary and consequently allocation of endoscopy time would be wrong. A slight overbooking might be needed to counterbalance this risk. In any case, accurate patient selection is always needed together with a full informed consent before the procedure. Lastly, unnecessary anticoagulant withdrawals may occur when the patients are prepared for the same session approach and ERCP ultimately turns out not to be indicated.

## 6. Overcoming ERCP Limitations with EUS

In addition to the good outcomes of the same session EUS-ERCP linear strategy (i.e., EUS first and then ERCP), it is sometimes necessary to return to EUS in case of ERCP failures and limitations.

EUS rendezvous is an alternative to interventional radiology-guided rendezvous after failed biliary cannulation that can be performed in the same endoscopic session (CP). A recent systematic review and proportion meta-analysis including 342 patients demonstrated an 86% pooled rate of technical success [27]. The pooled rate of clinical success was 81% and adverse events occurred in 14%. As usual, given the risk of adverse events, these advanced procedures are best suited in centers with high expertise in bilio-pancreatic therapeutic endoscopy.

Other limitations with ERCP can be encountered when cannulation fails due to periampullary diverticulum or post-surgical anatomical changes. Garcia–Alonso et al. reported 170 cases of salvage EUS-guided ductal access and drainage ERCP failures [28]. In approximately half of the cases, EUS-guided drainage procedures were performed anticipating rather than following ERCP failures (e.g., post-surgical anatomy). The overall rate of EUS salvage was 7.7% out of a cohort of >2000 ERCP procedures. Interestingly, this high EUS salvage rate reflected disease complexity, a wide definition of ERCP failure, and restrictive percutaneous drainage use, not poor ERCP skills. A case of EUS-guided choledocho-duodenostomy with electrocautery-enhanced lumen-apposing metal stent was also reported by the same group in a patient with benign biliary obstruction due to a large periampullary diverticulum [29].

Lastly, EUS-guided gallbladder drainage (GBD) can be coupled to ERCP in non-surgical candidates [30]. We owe to the same Spanish group of the studies above the description of same session ERCP and EUS-GBD as a strategy to comprehensively treat gallstone disease in selected patients [31]. The technical and clinical success rates of the combined procedures were comparable to EUS-GBD alone, indicating that it is possible to combine EUS-GBD and ERCP without increasing adverse events as a comprehensive treatment of gallstone disease by purely endoscopic means.

## 7. Conclusions

EUS and ERCP are highly advanced techniques that have come together to form a new endoscopic art, which is bilio-pancreatic endoscopy. Given the skills required to perform these demanding procedures, accurate planning of training and maintenance of competence are essential. In patients with suspected CBD stones, EUS is always useful to obtain all the diagnostic features that are functional to the appropriate treatment with ERCP. When performed within the same session, EUS and ERCP are mutually beneficial to obtain the best technical and clinical outcomes even in complicated cases, thereby minimizing adverse events and treatment failure.

## Figures and Tables

**Figure 1 medicina-57-01019-f001:**
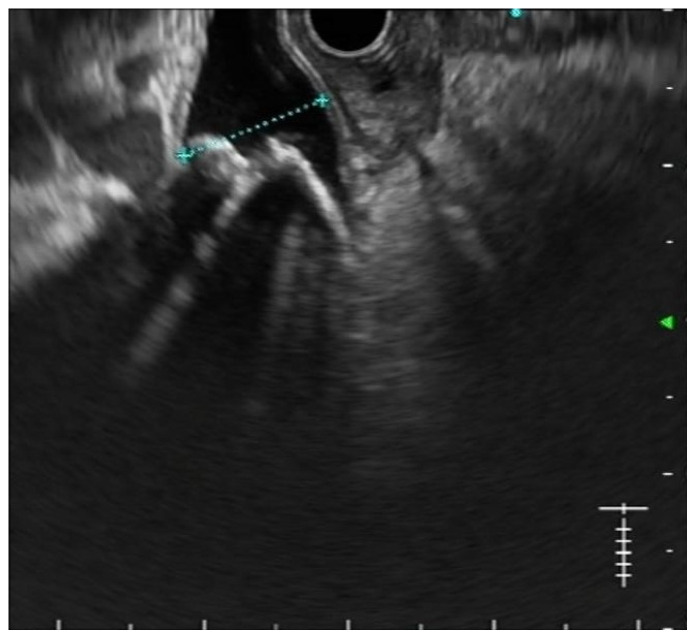
EUS shows multiple stones in a dilated common bile duct.

**Table 1 medicina-57-01019-t001:** Indications and benefits of same session EUS/ERCP.

Main Indications	Technical	Organizational	Clinical
Patients at high risk for CBD stones	Real time information from EUS	Reduction of the hospital length of stay	Reduced risk of cholangitis/acute pancreatitis while waiting after EUS
Symptomatic individuals with status post-cholecystectomy	Selecting the optimal approach to the papilla and CBD stone extraction	Reduction of hospital and endoscopy costs	Prompt treatment of cholangitis
Pregnant women	Alternative drainage in case of failed cannulation (e.g., rendezvous, EUS-GBD, EUS-CDS)	Cost-effective management of endoscopy unit time	Reduction of diagnostic ERCP due to migration of stones after EUS
Fragile patients unfit for surgery	Reduction in overall propofol dose	Combination with EUS-GBD for full treatment of biliary stones	Reduction of endoscopy-related adverse events

EUS: endoscopic ultrasound; ERCP: endoscopic retrograde cholangiopancreatography; CBD: common bile duct; GBD: gallbladder drainage; CDS: choledocho-duodenostomy.

## Data Availability

Not applicable.

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
