# Peer review of "EUS and ERCP in the Same Session for Biliary Stones: From Risk Stratification to Treatment Strategy in Different Clinical Conditions"

_medicina, 2021, doi:10.3390/medicina57101019_

Round 1
Reviewer 1 Report
The authors stated the clinical benefits of performing EUS and ERCP in the same session. It is a well-written and interesting article. However, there are some concerns about this article.
The authors described that the combined procedures' technical and clinical success rates were comparable to EUS-GBD alone. In addition, it is possible to combine EUS-GBD and ERCP without increasing adverse events. However, these procedures are not common in other countries, even in a high-volume center. How do you think? It mainly depends on the examiner's skills. Would you please state the criteria of skills of endoscopists (i.e., How many cases were experienced?)
In the manner of the review article, even narrative review like this article, the author should state the article selection procedure (i.e., Pubmed, Web of Science). Overall, this is like an Editorial, not a systemic review article.
Author Response
...How do you think? It mainly depends on the examiner's skills. Would you please state the criteria of skills of endoscopists (i.e., How many cases were experienced?)
Re. Thank you. I have reported the ESGE curriculum in order to address this issue. I have added a paragraph reporting the number of procedures that are needed in EUS and ERCP in order to demonstrate competence.
...the author should state the article selection procedure.
Re. This is an expert narrative review. There has been neither a systematic review nor a meta-analysis of the literature. All the information reported in the text and the relative references came from the authors' experience and knowledge.
Reviewer 2 Report
This review article well summarized the advantages of same session EUS-ERCP. There are only a couple of minor things to be addressed.
- In line 57-58, I cannot agree that EUS is associated with the highest risk of serious complications and mortality. The interventional EUS can cause serious adverse events, but diagnostic EUS dose not.
- Please summarize the potential indication of the same session EUS-ERCP linear strategy in choledocholithiasis with Table.
- For the same session EUS-ERCP for choledocholithiasis, which is preferred, radial EUS or linear EUS?
Author Response
...In line 57-58, I cannot agree that EUS is associated with the highest risk of serious complications and mortality. The interventional EUS can cause serious adverse events, but diagnostic EUS dose not.
Re. I agree. I have added the term "interventional" to make it clearer
---summarize the potential indication of the same session EUS-ERCP linear strategy in choledocholithiasis with Table.
Re. Done (added a column to table 1)
...For the same session EUS-ERCP for choledocholithiasis, which is preferred, radial EUS or linear EUS?
Re. According to the literature, both radial and linear EUS are accurate for the diagnosis of CBD stones. I have added a sentence to the text.
Round 2
Reviewer 1 Report
1)There are no descriptions of primary outcomes at the end of the introduction. Would you state like " In this narrative reviwe~"?
2) What is space line46-47 for?
I enjoyed reading your article.
Author Response
I have added a statement at the end of the introduction to describe this as a narrative review.
I have removed the space. I think it was just some formatting issue.
Thank you very much for your positive comments!